# Simulated sunlight decreases the viability of SARS-CoV-2 in mucus

**Angela Sloan**[1]*, **Todd Cutts**[1], **Bryan D. Griffin**[1¤a], **Samantha Kasloff**[1],
**Zachary Schiffman**[1,2], **Mable Chan**[1], **Jonathan Audet**[1], **Anders Leung**[1], **Darwyn Kobasa**[1,2],
**Derek R. Stein**[1¤b], **David Safronetz**[1,2], **Guillaume Poliquin**[1,3]*

**1** National Microbiology Laboratory, Public Health Agency of Canada, Winnipeg, Manitoba, Canada,
**2** Department of Medical Microbiology and Infectious Diseases, University of Manitoba, Winnipeg, Manitoba,
Canada, **3** Department of Pediatrics and Child Health, College of Medicine, Faculty of Health Sciences,
University of Manitoba, Winnipeg, Manitoba, Canada

¤a Current address: Biological Sciences, Sunnybrook Research Institute, Toronto, Ontario, Canada
¤b Current address: Cadham Provincial Laboratory, Winnipeg, Manitoba, Canada
* guillaume.poliquin@canada.ca (GP); angela.sloan@canada.ca (AS)

**Data Availability Statement:** All relevant data are within the manuscript and its Supporting information files.

**Funding:** The authors received no specific funding for this work.

## Abstract

The novel coronavirus, SARS-CoV-2, has spread into a pandemic since its emergence in Wuhan, China in December of 2019. This has been facilitated by its high transmissibility within the human population and its ability to remain viable on inanimate surfaces for an extended period. To address the latter, we examined the effect of simulated sunlight on the viability of SARS-CoV-2 spiked into tissue culture medium or mucus. The study revealed that inactivation took 37 minutes in medium and 107 minutes in mucus. These times-to-inactivation were unexpected since they are longer than have been observed in other studies. From this work, we demonstrate that sunlight represents an effective decontamination method but the speed of decontamination is variable based on the underlying matrix. This information has an important impact on the development of infection prevention and control protocols to reduce the spread of this deadly pathogen.

## Introduction

Severe acute respiratory syndrome coronavirus 2 (SARS-CoV-2) has been responsible for a global pandemic of its associated disease, COVID-19, since its emergence in Wuhan, China in December of 2019 [1–5]. Efforts to contain the spread of the virus have required whole-of-society mobilization efforts, such as physical distancing and business closures, which have led to worldwide economic devastation. The need for such measures has been driven in part by the ability of this virus to be transmitted by asymptomatic carriers and pre-symptomatic patients [6–8]. The role of fomites may also be important, as the virus can remain infectious on non-porous surfaces from 3 to 28 days depending on ambient conditions [9–11].

Natural sunlight harbors three types of ultraviolet (UV) radiation: UVA (315–400 nm) which is almost completely absorbed by the Earth's atmosphere, UVB (280–315 nm) which is partially absorbed, and UVC (100–280 nm) which reaches the Earth [12]. While UVC radiation is considered germicidal and has been shown to deactivate microorganisms, including

**Competing interests:** The authors have declared that no competing interests exist.

SARS-CoV-2, in a variety of settings [13–19], radiation in the UVA and UVB spectral ranges are also known to cause significant damage to nucleic acids [20]. These observations suggest that sunlight penetrating the atmosphere may also act as an antimicrobial agent. Indeed, monthly average percent positive infection rates of SARS-CoV-2 have been negatively correlated with sunlight UV radiation dose [21]. Moreover, recent studies have examined the ability of sunlight to inactivate SARS-CoV-2 in simulated saliva, with intriguing results [22]. Ratnesar-Shumate et al. have shown that 90% of live SARS-CoV-2 is deactivated every 6.8 to 12.8 minutes depending on the integrated UVB irradiance [22]. In this study, we examine the ability of sunlight to deactivate SARS-CoV-2 in simulated mucus at a level of UVB irradiance not previously tested. These data will add to the growing field of knowledge concerning the ability of sunlight to act as a natural sterilizing medium and reduce the viability of SARS-CoV-2.

## Materials and methods

### Virus propagation

The initial virus aliquot of SARS-CoV-2 (cultured from patient sample; viral passage 1; hCoV-19/Canada/ON-VIDO-01/2020, GISAID accession# EPI_ISL_425177) was provided by the Vaccine and Infectious Disease Organization (VIDO; Saskatoon, Saskatchewan, Canada). Vero E6 (ATCC® CRL-1586™) cells were grown in 150 cm$^2$ tissue-cultured treated flasks to 80–90% confluence in Dulbeco's Minimum Essential Medium (DMEM) supplemented with 5% bovine calf serum (BCS). Within a biosafety level (BSL)-3 laboratory, the medium was removed and the cells were washed with DMEM containing 0.1% bovine serum albumin (BSA). The cells were then infected with the SARS-CoV-2 aliquot (5 μL) in DMEM (5 mL) containing 0.5 μg/mL TPCK-Trypsin and 0.1% BSA and incubated at 37°C and 5% $CO_2$. After 30 min of absorption with intermittent rocking every 5–10 min, additional maintenance medium (30 mL) was added and the cells were again incubated at 37°C and 5% $CO_2$. Any resulting cytopathic effect (CPE) was monitored daily, with the supernatant harvested 5 days post-infection (dpi). The initial virus inoculum was quantified by end-point titration on Vero E6 cells and determined to be $4.6 \times 10^6$ TCID$_{50}$/mL (i.e. 50% tissue culture infectious dose per milliliter).

### Organic matrix

The organic matrix used in this study is the standard tripartite soil load described in ASTM E2197-17 e1 [23]. Exceptionally, the mucin suspension in 0.85% NaCl was gamma-irradiated at 2 megarads on wet ice as an alternative to filter-sterilization to avoid clogging of the filter.

### Solar simulator

The artificial sunlight used in this study was produced by the SunLite Solar Simulator Model 11002 from Abet Technologies. Solar output was set to 1 sun, defined as full sunlight intensity on a bright clear day on Earth and measuring approximately 1000 watts per square meter [24]. The spectral data provided by Abet Technologies for our particular unit measured the UVA and UVB output at 1 sun as 41.46 watts per square meter (W/m$^2$) UVA and 1.28 W/m$^2$, respectively, approximately equivalent to the irradiance measured at noon on the 2020 spring and fall equinoxes (March 21 and September 22, respectively) at 40°N latitude and sea level (National Center for Atmospheric Research Tropospheric Ultraviolet and Visible Radiation Model). An atmospheric edge filter was used to block all wavelengths below 300 nm, as radiation below this level is absorbed by the atmosphere in the natural environment.

## Coupon tests

Although SARS-CoV-2 was initially propagated in Vero E6 cells (described above), subsequent cell culture assays were performed in Vero cells, as a more evident CPE was observed in the latter cell line. Vero cells (ATCC® CCL-81™) were seeded into clear, flat-bottomed, tissue-culture treated 96-well plates in Minimum Essential Medium (MEM; 100 μL) supplemented with 5% BCS and 1% L-glutamine, and grown to approximately 90% confluence overnight at 37˚C and 5% $CO_2$. All subsequent procedures were performed in a BSL-4 laboratory in a class II biological safety cabinet by workers according to approved internal procedures. One stock vial of SARS-CoV-2 was thawed at room temperature and 340 μL added to 160 μL of either maintenance medium (MEM containing 1% BCS and 1% L-glutamine) which emulates the virus in a laboratory environment, or organic matrix (described above) which emulates the virus in its natural environment [25]. Positive controls were prepared in triplicate by adding the viral suspension (10 μL) to maintenance medium (1 mL). Coupons were prepared by adding the viral suspension (10 μL) to the centre of sterilized stainless steel disks (1 cm in diameter and 0.7 mm thick) and allowed to dry for 45 min. Coupons were prepared in triplicate for test and control conditions at each designated time-point. Once dry (i.e. time-point "0"), maintenance medium (1 mL) was pipetted up and down on each of 3 coupons to re-suspend the virus and used to infect Vero cells (see below). In experiments designed to control for the confounding variable of heat, half of the coupons were placed directly under the solar simulator light source set to 1 sun in a digital block heater/cooler set to 14˚C, which ensured the coupons remained at room temperature (22.5˚C). Corresponding coupons were placed in a petri dish within the BSC. For experiments where infrared heat was permitted, control coupons were placed in the block heater/cooler and heated to the same temperature the disks reached under the solar simulator, which was periodically measured with a thermometer and wire probe. Relative humidity (RH) was measured with a hygrometer (TFA KlimaLogg Pro).

Shortly after collection, each sample and positive control were used to infect the nearly confluent Vero cells in triplicate: after removal of the growth medium, neat virus (100 μL) was added to the top row of the 96-well plate and a series of virus dilutions ($10^{-1}$ to $10^{-6}$) prepared in maintenance medium were added to the 6 consecutive rows below. The wells in the bottom row of the plate contained maintenance medium only and served as negative controls. The plates were incubated at 37˚C and 5% $CO_2$ for 4 days, at which time individual wells were examined for cytopathic effect (CPE).

## Graphing and statistics

The Reed and Muench calculation [26] was used to calculate $TCID_{50}$/mL values and the limit of quantitation. $Log_{10}$ $TCID_{50}$/mL values were then calculated and plotted to compare virus viability in all treatment groups over time. Viral titers were fit to curves using linear regression in GraphPad Prism (GraphPad Software, www.graphpad.com.), with slopes representing the rate of viral inactivation. D'Agostino-Pearson normality tests using a straight line equation were used to confirm Guassian residuals for all data sets. Runs tests were performed to confirm the assumption of linearity for all regression lines. Values below the limit of quantitation were not used in the construction of regression lines or in statistical analyses. All statistical analyses were performed using GraphPad Prism.

## Results and discussion

The top four graphs in Fig 1 demonstrate that the viability of SARS-CoV-2 decreased significantly faster when exposed to sunlight versus exposure to the same conditions with sunlight removed (p ≤ 0.0045). The bottom two graphs show the relationship between coupon

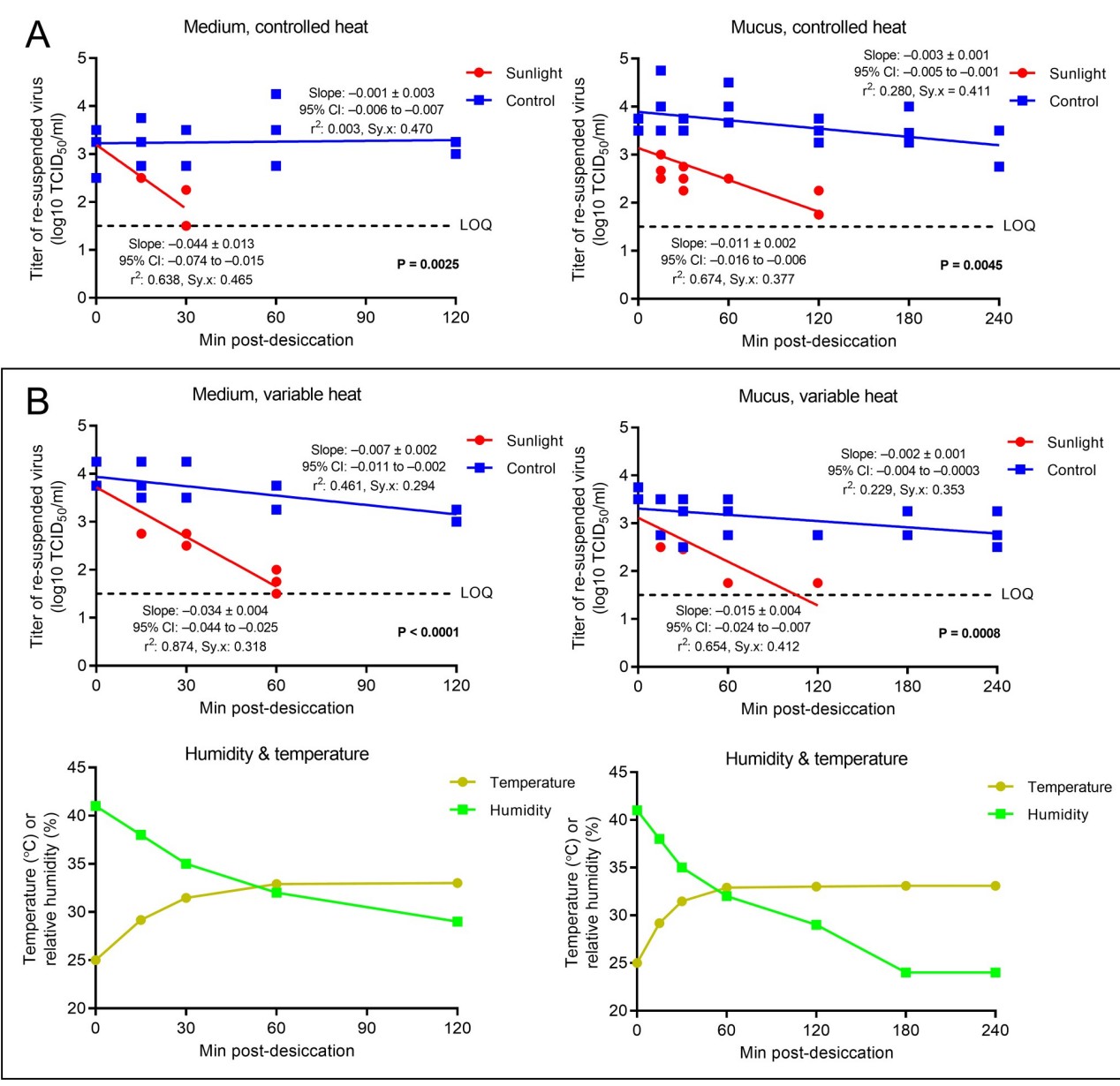

**Fig 1. Viability of SARS-CoV-2 on stainless steel after exposure to simulated sunlight.** SARS-CoV-2 was suspended in (A) culture medium or (B) an organic matrix, deposited on stainless steel, desiccated, and exposed to either simulated sunlight ("Sunlight") or corresponding ambient conditions ("Control"). Graphs in (A) and the top row of (B) show linear regression fits for the titer of viable eluted virus, expressed as the $\log_{10}$ 50% tissue culture infectious dose per milliliter ($TCID_{50}$/mL), following culture in Vero cells. The limit of quantitation (LOQ), denoted by a dashed line, is 1.5 logs or 3.16 x $10^1$ $TCID_{50}$/mL. Plots show 3 biological replicates per time-point, with each biological replicate representing the average of 3 technical replicates. Viability values falling below the limit of quantitation were not quantifiable and therefore excluded from analysis. Graphs in (A) represent heat-controlled experiments where coupon temperature was held constant at 22.5˚C (relative humidity was 34%). Graphs in the bottom row of (B) show coupon temperature and relative humidity readings measured at each time-point during heat-permitted assays, and correspond to the experiment represented in the graph located directly above. The viability of SARS-CoV-2 observed after sunlight exposure was observed to be significantly lower than that measured in control experiments under all conditions (p ≤ 0.0045). The slope of the regression line for control virus diluted in medium under heat-controlled conditions was not significantly different from zero (p = 0.849). The slope, 95% CI, and goodness of fit parameters, $r^2$ (coefficient of determination) and Sy.x (SE of estimate), for each regression line are shown above and below the best-fit curves representing the control and experimental conditions, respectively. Raw data and the results of additional statistical analyses are shown in S1–S4 Tables. All graphs were created and statistical analyses performed using GraphPad Prism software (GraphPad Software, www.graphpad.com).

**Table 1. Results of linear regression analyses between data sets.**

| Variable | | | F-value | df | p-value | Significant? |
|---|---|---|---|---|---|---|
| Condition | Matrix | Heat Type | | | | |
| Sun/Control | Medium | V | 38.6374 | 22 | <0.0001 | Extremely |
| Sun/Control | Mucus | V | 14.0614 | 28 | 0.0008 | Extremely |
| Sun/Control | Medium | C | 11.8869 | 20 | 0.0025 | Very |
| Sun/Control | Mucus | C | 9.4114 | 30 | 0.0045 | Very |
| Sun | Medium/Mucus | V | 11.0209 | 19 | 0.0036 | Very |
| Sun | Medium/Mucus | C | 8.3886 | 18 | 0.0096 | Very |
| Sun | Mucus | V/C | 0.9713 | 20 | 0.3361 | No |
| Sun | Medium | V/C | 0.7406 | 17 | 0.4014 | No |
| Control | Medium/Mucus | V | 3.1546 | 31 | 0.8553 | No |
| Control | Medium/Mucus | C | 1.4304 | 32 | 0.2405 | No |
| Control | Mucus | V/C | 0.2634 | 38 | 0.6107 | No |
| Control | Medium | V/C | 3.7758 | 25 | 0.0633 | No |

df, degrees of freedom; V, variable; C, controlled. Linear regression analyses were performed to determine statistical significance between the slopes of regression lines in compared data sets.

temperature and RH in experiments where heat was allowed to rise with sunlight exposure, and correspond to the experiment represented in the graph located directly above. Viral titer decreased most rapidly when the virus was suspended in culture medium under controlled heat (Fig 1A, left graph), from 3.12 to 1.5 $\log_{10}$ $TCID_{50}$/mL in 37 minutes at 22.5°C and 34% RH. Although not statistically significant, the same reduction in viability took 47 minutes during heat- and RH-variable assays (Fig 1B, top-left graph; Table 1, p = 0.4014). The presence of an organic matrix significantly extended the survival of the virus when exposed to sunlight. In these trials, viral viability was reduced more slowly when the virus was suspended in simulated mucus than in medium for both heat-controlled (Fig 1A, right graph; Table 1, p = 0.0096) and heat-variable (Fig 1B, top-right graph; Table 1, p = 0.0036) assays. In mucus, sunlight exposure for 147 minutes was necessary to reduce virus titer from 3.12 to 1.5 $\log_{10}$ $TCID_{50}$/mL at room temperature (Fig 1A, right graph). This was reduced to 107 minutes when heat and RH were variable (Fig 1B, top-right graph), but the difference was not statistically significant (Table 1, p = 0.3361). Raw data and details of the additional statistical analyses performed are shown in S1–S4 Tables.

The present study demonstrates that 1.28 W/m$^2$ of UVB radiation is capable of inactivating SARS-CoV-2 desiccated on stainless steel surfaces. Further, the matrix within which the virus is suspended has a demonstrable impact on the effect of sunlight as a disinfection agent. Inactivation of a 3.12 log inoculum occurred faster when simple media was used (37–47 minutes) compared to simulated mucus (107–147 minutes). We also observed much slower viral decay rates compared to a recent report by Ratnesar-Shumate et al., who demonstrated medium-suspended SARS-CoV-2 being inactivated at a rate almost twice that seen in our investigation [22]. The former study examined the effect of sunlight at three different UVB irradiances, 0.3, 0.68, and 1.63 W/m$^2$, on the decay rate of SARS-CoV-2 over time, while we assessed UVB irradiance at 1.28 W/m$^2$. Plotting the rate of viral decay (i.e. log $TCID_{50}$/min) of medium-suspended SARS-CoV-2 against the three examined UVB irradiances, an expected slope value of -0.063 was extrapolated at 1.28 W/m$^2$; however, ours was almost half that at -0.034 (S1 Fig). Moreover, when the virus was spiked into simulated saliva in the aforementioned study, the rate of viral inactivation was almost nine times faster than the inactivation rate we observed in

mucus. Plotting the same parameters obtained for saliva-suspended virus, the expected slope value of the three UVB irradiances was -0.136, whereas the slope value obtained for mucus-suspended virus was -0.015, over nine times lower than that observed for saliva (S1 Fig). Interestingly, the former study also demonstrated accelerated inactivation in simulated saliva when compared to tissue culture media. The reason for this is unclear, since accelerated inactivation was not observed in simulated saliva in the absence of sunlight. From these findings and our results, it appears that sunlight and matrices can have complex interactions. Further, simulated mucus appears to be protective to SARS-CoV-2, perhaps due to the presence of one or more of the three types of protein (high molecular weight proteins, low molecular weight peptides, and mucus material) [23]. Lastly, our results did not demonstrate a significant difference in inactivation time based on changes in temperature and RH, which stands in contrast to data from Biryukov et al., who showed that SARS-CoV-2 naturally decays more rapidly with an increase in either temperature or RH [27]. It is possible that, in the current study, any increase in the viral decay rate due to rising coupon temperature was offset by decreasing relative humidity and vice versa.

Our findings are important in demonstrating that sunlight can be used to decontaminate surfaces confirmed or suspected of having been exposed to SARS-CoV-2. However, our study has important limitations. First, we examined sunlight conditions equivalent to the irradiance measured at noon on the 2020 spring and fall equinoxes (March 21 and September 22, respectively) at 40˚N latitude and sea level. As solar intensity varies geographically, it would be important to adjust exposure times to deliver a similar solar radiation dose based on local conditions. A second limitation is the use of a non-porous surface for these experiments. Stainless steel was specifically selected as a representative non-porous surface, as our study aimed to reproduce the standard quantitative disk carrier test method described by the ASTM E2197. It is known that surface characteristics can impact survival of the virus, with non-permeable surfaces allowing the virus to persist longer than do absorbent materials [9, 10, 28, 29]. It is possible that testing porous surfaces, such as wood or concrete, would result in much faster viral decay rates. Finally, we examined simulated mucus but no other spiked or simulated bodily fluids. SARS-CoV-2 RNA has been detected routinely from patients, but recovering infectious virus appears to be much less frequent [30, 31]. In spite of this, it would be worthwhile to examine the effect of sunlight on SARS-CoV-2 in other matrices (e.g. naso-/oro-pharyngeal fluids, stool, etc.) where infectious virus has been recovered [30, 31].

Overall, our study provides important information on the ability of sunlight to decontaminate surfaces exposed to mucus-suspended SARS-CoV-2. Notably, the rate of viral deactivation is significantly longer than that previously determined for SARS-CoV-2 suspended in saliva. These findings are important for determining plans for the maintenance and decontamination of outdoor spaces as public health measures are relaxed. Next steps should include examining the effect of sunlight on SARS-CoV-2 in other matrices where infectious virus has been recovered.

## Supporting information

**S1 Fig. Viral decay rate of medium- and saliva-suspended SARS-CoV-2 according to UVB exposure.** By plotting the viral decay rates of (A) medium- and (B) saliva-suspended SARS-CoV-2 observed by Ratnesar-Shumate et al. [22] at different UVB intensities, expected decay rates for medium- and mucus-suspended SARS-CoV-2 were extrapolated for the UVB intensity examined in the current study. In line with Ratnesar-Shumate et al.'s findings, the viral decay of medium-suspended virus (A) at a UVB intensity of 1.28 watts per square meter (W/$m^2$) should have been -0.063 $\log_{10}$ $TCID_{50}$/min; however, our results show it occurred at

-0.034 $\log_{10}$ $TCID_{50}$/min. Comparing bodily matrices (B), according to Ratnesar-Shumate et al., saliva-suspended SARS-CoV-2 should decay at a rate of -0.136 $\log_{10}$ $TCID_{50}$/min when UVB equals 1.28 W/m$^2$. The current study shows that mucus-suspended SARS-CoV-2 decays at a rate of over nine times slower, at 0.015 $\log_{10}$ $TCID_{50}$/min.
(TIF)

**S1 Table. Log10 $TCID_{50}$/mL values used to create linear regression lines for each data set.**
(DOCX)

**S2 Table. Best fit and goodness of fit parameters for linear regression lines constructed for each data set.**
(DOCX)

**S3 Table. Results of D'Agostino-Pearson normality tests used to confirm Guassian residuals for each data set.**
(DOCX)

**S4 Table. Results of runs tests used to confirm the assumption of linearity for each regression line.**
(DOCX)

## Acknowledgments

We thank the Vaccine and Infectious Diseases Organization (VIDO; Saskatoon, Canada) for providing a Vero cell culture passage 1 isolate of SARS-CoV-2 (hCoV-19/Canada/ON-VIDO-01/2020, GISAID accession# EPI_ISL_425177). We also thank Abet Technologies for their provision of technical expertise.

## Author Contributions

**Conceptualization:** Guillaume Poliquin.

**Formal analysis:** Angela Sloan.

**Investigation:** Angela Sloan, Todd Cutts, Bryan D. Griffin, Anders Leung, Darwyn Kobasa, Guillaume Poliquin.

**Methodology:** Angela Sloan, Todd Cutts, Derek R. Stein.

**Project administration:** Angela Sloan, Guillaume Poliquin.

**Resources:** Angela Sloan, Samantha Kasloff, Zachary Schiffman, Mable Chan, Guillaume Poliquin.

**Supervision:** David Safronetz, Guillaume Poliquin.

**Validation:** Angela Sloan.

**Visualization:** Angela Sloan, Zachary Schiffman, Jonathan Audet.

**Writing – original draft:** Angela Sloan.

**Writing – review & editing:** Todd Cutts, Bryan D. Griffin, Samantha Kasloff, Zachary Schiffman, Mable Chan, Jonathan Audet, Anders Leung, Darwyn Kobasa, Derek R. Stein, David Safronetz, Guillaume Poliquin.

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
