## [Decision Letter · Decision Letter 0]

26 Mar 2021

PONE-D-21-01351

Simulated sunlight decreases the viability of SARS-CoV-2 in mucus

PLOS ONE

Dear Dr. Sloan,

Thank you for submitting your manuscript to PLOS ONE. After careful consideration, we feel that it has merit but does not fully meet PLOS ONE’s publication criteria as it currently stands. Therefore, we invite you to submit a revised version of the manuscript that addresses the points raised during the review process.

Please attend to all the comments and concerns that have been raised by the reviewers. Ensure that you have given adequate background (introduction) to your work and that the results have been adequately discussed. You also need to improve the quality of your figures.

We look forward to receiving your revised manuscript.

Kind regards,

Martin Chtolongo Simuunza, PhD

Academic Editor

PLOS ONE

Journal Requirements:

Reviewers' comments:

Reviewer's Responses to Questions

**Comments to the Author**

1. Is the manuscript technically sound, and do the data support the conclusions?

Reviewer #1: Partly

Reviewer #2: Yes

2. Has the statistical analysis been performed appropriately and rigorously? 

Reviewer #1: Yes

Reviewer #2: Yes

3. Have the authors made all data underlying the findings in their manuscript fully available?

Reviewer #1: Yes

Reviewer #2: Yes

4. Is the manuscript presented in an intelligible fashion and written in standard English?

Reviewer #1: Yes

Reviewer #2: Yes

5. Review Comments to the Author

Reviewer #1: The COVID-19 outbreak is an ongoing global pandemic caused by severe acute respiratory syndrome coronavirus 2 (SARS-CoV-2). However, SARS-CoV-2 is less deadly but far more transmissible than other coronaviruses, such as MERS-CoV or SARS-CoV. The SARS-CoV-2 viruses remain viable and stable on various environmental conditions or surfaces for an extended period. In the current study, the authors aimed to assess the impact of stimulated sunlight on the viability of SARS-CoV-2. The study could provide useful information for pandemic mitigation efforts.

Major Issues:

1. It is unclear why sunlight would be considered a natural sterilizing medium. The study hypothesis and study plan need to be emphasized to highlight the importance of the current study. A more detailed background literature research should be reported as to how the current study can improve our understanding of SARS-CoV-2 from the already reported studies.

2. Sunlight or the sun’s angle is affected by geographical variations, weather condition and season. As reported previously by Ratnesar-Shumate et al. (2020), three different integrated UVB irradiance levels represented the spectra utilized span UVB irradiances throughout the different season. Could the authors explain why the solar output for spring and fall was selected for the current study?

3. For any reported results in the text, authors should refer the results to the exact figure. For example, “Fig 1 demonstrates that the viability of SARS-CoV-2 decreased significantly faster when exposed to sunlight versus exposure to the same conditions with sunlight removed (p ≤ 0.0045). Viral titer decreased most rapidly when the virus was suspended in the culture medium, from 3.12 to 1.5 log10 TCID50/mL in 37 minutes at 22.5°C and 34% RH.” It is unclear which graphs are this reported information belongs. There are six graphs in figure 1, and perhaps a more detailed explanation for each graph or condition would be beneficial for readers. Authors could consider reporting a summarized version of the supplementary data if they reported it multiple times in the text.

4. The discussion was vaguely written; the authors need to provide a direct comparison with other previous studies. It is also unclear what other additional information that the current study has provided to the field. A paragraph of limitation should be provided to the current study.

Minor Issues:

1. The figure quality needs to be improved significantly for publishable quality. Information on the figures appeared very blurry, and it was not easy to follow or read.

2. Abbreviation such as RH needs to be clearly stated in the text.

Reviewer #2: The manuscript by Sloan et al describes the evaluation of SARS-CoV-2 inactivation by artificial sunlight in the context of cell culture medium or a standardized mucous matrix. These studies showed that the sunlight system used in these studies effectively inactivated SARS-CoV-2 in the test system and also showed that the mucous matrix reduced the efficiency of inactivation.

Minor comment:

Line 16: Log10 (subscript)

6. PLOS authors have the option to publish the peer review history of their article (what does this mean?). If published, this will include your full peer review and any attached files.

Reviewer #1: No

Reviewer #2: No

---

## [Author Response · Author response to Decision Letter 0]

5 May 2021

Reviewer's Responses to Questions

Comments to the Author

1. Is the manuscript technically sound, and do the data support the conclusions?

Reviewer #1: Partly

Reviewer #2: Yes

2. Has the statistical analysis been performed appropriately and rigorously? 

Reviewer #1: Yes

Reviewer #2: Yes

3. Have the authors made all data underlying the findings in their manuscript fully available?

Reviewer #1: Yes

Reviewer #2: Yes

4. Is the manuscript presented in an intelligible fashion and written in standard English?

Reviewer #1: Yes

Reviewer #2: Yes

5. Review Comments to the Author

Reviewer #1: The COVID-19 outbreak is an ongoing global pandemic caused by severe acute respiratory syndrome coronavirus 2 (SARS-CoV-2). However, SARS-CoV-2 is less deadly but far more transmissible than other coronaviruses, such as MERS-CoV or SARS-CoV. The SARS-CoV-2 viruses remain viable and stable on various environmental conditions or surfaces for an extended period. In the current study, the authors aimed to assess the impact of stimulated sunlight on the viability of SARS-CoV-2. The study could provide useful information for pandemic mitigation efforts.

Major Issues:

1. It is unclear why sunlight would be considered a natural sterilizing medium. The study hypothesis and study plan need to be emphasized to highlight the importance of the current study. A more detailed background literature research should be reported as to how the current study can improve our understanding of SARS-CoV-2 from the already reported studies.

Response: A paragraph has been added to the Introduction explaining why the UV radiation present in sunlight can be used as a natural sterilizing mechanism to deactivate microorganisms, including SARS-CoV-2. We have also referenced other studies that have used UV radiation and sunlight to deactivate SARS-CoV-2 and described how the findings of our research will improve our understanding of the virus.

2. Sunlight or the sun’s angle is affected by geographical variations, weather condition and season. As reported previously by Ratnesar-Shumate et al. (2020), three different integrated UVB irradiance levels represented the spectra utilized span UVB irradiances throughout the different season. Could the authors explain why the solar output for spring and fall was selected for the current study?

Response: We chose to examine the ability of simulated sunlight to deactivate SARS-CoV-2 using our solar simulator set to “1 sun”, which is defined as full sunlight intensity on a bright clear day on Earth and measuring approximately 1000 W/m2 [Tables for Reference Solar Spectral Irradiances: Direct Normal and Hemispherical on 37 Tilted Surface". 2008. doi:10.1520/G0173-03R08]. At this setting, the UVB emitted by our particular solar simulator produced a UVB irradiance of 1.28 W/m2 at 1 sun, which happened to be equivalent to the amount of UVB emitted during the day on both the spring solstice and autumn equinox. As this level of UVB represented the irradiance emitted on separate days during two distinct seasons, 1 sun was considered an ideal level of solar output to examine. We also felt that less intense solar outputs (i.e. less than what would normally be observed during summer) resulting in prolonged viral sustainability would have more real-world implications. 

3. For any reported results in the text, authors should refer the results to the exact figure. For example, “Fig 1 demonstrates that the viability of SARS-CoV-2 decreased significantly faster when exposed to sunlight versus exposure to the same conditions with sunlight removed (p ≤ 0.0045). Viral titer decreased most rapidly when the virus was suspended in the culture medium, from 3.12 to 1.5 log10 TCID50/mL in 37 minutes at 22.5°C and 34% RH.” It is unclear which graphs are this reported information belongs. There are six graphs in figure 1, and perhaps a more detailed explanation for each graph or condition would be beneficial for readers. Authors could consider reporting a summarized version of the supplementary data if they reported it multiple times in the text.

Response: Each graph in Figure 1 has been described and referenced more clearly. Supplementary Table 2 has become Table 1, as it was referenced several times in the Results section.

4. The discussion was vaguely written; the authors need to provide a direct comparison with other previous studies. It is also unclear what other additional information that the current study has provided to the field. A paragraph of limitation should be provided to the current study.

Response: The Discussion has been rewritten to include a more direct comparison with the only other study reported to date examining simulated sunlight on the decay rate of SARS-CoV-2 on surfaces (i.e. Ratnesar-Shumate et al. 2020). A supplementary figure has also been included to directly compare the two studies’ results. Lastly, a paragraph discussing limitations has been added.

Minor Issues:

1. The figure quality needs to be improved significantly for publishable quality. Information on the figures appeared very blurry, and it was not easy to follow or read.

Response: We have submitted our original figure to the Preflight Analysis and Conversion Engine (PACE) digital diagnostic tool at https://pacev2.apexcovantage.com/increased. To the best of our knowledge, the submitted figure should now conform to PLOS ONE standards. Unfortunately, while the PACE-created Fig 1 is extremely clear in its submitted .tif file format, it appears blurry in the final PDF submission. It is our hope that we can work together with the journal editors if the quality of Fig 1 continues to be a problem.

2. Abbreviation such as RH needs to be clearly stated in the text.

Response: The “RH” abbreviation has been defined as “relative humidity” in the Materials and Methods section. We have also defined the abbreviation “W/m2” as “watts per square meter” and have spelled out “MRads” as “megarads”.

Reviewer #2: The manuscript by Sloan et al describes the evaluation of SARS-CoV-2 inactivation by artificial sunlight in the context of cell culture medium or a standardized mucous matrix. These studies showed that the sunlight system used in these studies effectively inactivated SARS-CoV-2 in the test system and also showed that the mucous matrix reduced the efficiency of inactivation.

Minor comment:

Line 16: Log10 (subscript)

Response: The subscript has been included.

6. PLOS authors have the option to publish the peer review history of their article (what does this mean?). If published, this will include your full peer review and any attached files.

Do you want your identity to be public for this peer review? For information about this choice, including consent withdrawal, please see our Privacy Policy.

Reviewer #1: No

Reviewer #2: No

---

## [Decision Letter · Decision Letter 1]

28 May 2021

Simulated sunlight decreases the viability of SARS-CoV-2 in mucus

PONE-D-21-01351R1

Dear Dr. Sloan,

We’re pleased to inform you that your manuscript has been judged scientifically suitable for publication and will be formally accepted for publication once it meets all outstanding technical requirements.

Kind regards,

Martin Chtolongo Simuunza, PhD

Academic Editor

PLOS ONE

Additional Editor Comments (optional):

Reviewers' comments:

Reviewer's Responses to Questions

**Comments to the Author**

1. If the authors have adequately addressed your comments raised in a previous round of review and you feel that this manuscript is now acceptable for publication, you may indicate that here to bypass the “Comments to the Author” section, enter your conflict of interest statement in the “Confidential to Editor” section, and submit your "Accept" recommendation.

Reviewer #1: All comments have been addressed

Reviewer #2: All comments have been addressed

2. Is the manuscript technically sound, and do the data support the conclusions?

Reviewer #1: Yes

Reviewer #2: Yes

3. Has the statistical analysis been performed appropriately and rigorously? 

Reviewer #1: Yes

Reviewer #2: Yes

4. Have the authors made all data underlying the findings in their manuscript fully available?

Reviewer #1: Yes

Reviewer #2: Yes

5. Is the manuscript presented in an intelligible fashion and written in standard English?

Reviewer #1: Yes

Reviewer #2: Yes

6. Review Comments to the Author

Reviewer #1: The only concern is the result section which appeared to be relatively short and might need to be elaborate further.

Reviewer #2: No additional comments

Extra characters to meet the 100 character minimum...……………………………………………………...

7. PLOS authors have the option to publish the peer review history of their article (what does this mean?). If published, this will include your full peer review and any attached files.

Reviewer #1: No

Reviewer #2: No

---

## [Editor Report · Acceptance letter]

1 Jun 2021

PONE-D-21-01351R1 

Simulated sunlight decreases the viability of SARS-CoV-2 in mucus 

Dear Dr. Sloan:

I'm pleased to inform you that your manuscript has been deemed suitable for publication in PLOS ONE. Congratulations! Your manuscript is now with our production department. 

Kind regards, 

on behalf of

Dr. Martin Chtolongo Simuunza 

Academic Editor

PLOS ONE